# Rhizosphere Microbial Community Shows a Greater Response Than Soil Properties to Tea (*Camellia sinensis* L.) Cultivars

**Sirou Wei** [1,2,†], **Boheng Liu** [2,†], **Kang Ni** [2], **Lifeng Ma** [2,3], **Yuanzhi Shi** [2], **Yang Leng** [4], **Shenghong Zheng** [2], **Shuilian Gao** [1,5,*], **Xiangde Yang** [2,*] and **Jianyun Ruan** [2]

1   College of Horticulture, Fujian Agriculture and Forestry University, Fuzhou 350002, China
2   Tea Research Institute, Chinese Academy of Agriculture Sciences, Key Laboratory of Tea Biology and Resource Utilization of Tea, the Ministry of Agriculture, Hangzhou 310008, China
3   Xihu National Agricultural Experimental Station for Soil Quality, Hangzhou 310008, China
4   National Agricultural Technology Extension and Service Center, Ministry of Agriculture and Rural Affairs, Beijing 100125, China
5   Anxi College of Tea Science, Fujian Agriculture and Forestry University, Quanzhou 362406, China
*   Correspondence: gaoshuilian@126.com (S.G.); yangxd@tricaas.com (X.Y.)
†   These authors contributed equally to this work.

**Abstract:** Rhizosphere microbes play pivotal roles in regulating the soil ecosystem by influencing and directly participating in the nutrient cycle. Evidence shows that the rhizosphere microbes are highly dependent on plant genotype and cultivars; however, their characteristics in soils with different tea (*Camellia sinensis* L.) cultivars are poorly understood. Therefore, the present study investigated the rhizosphere soil properties, microbial community composition, and their potential functions under four tea cultivars Huangjinya (HJY), Tieguanyin (TGY), Zhongcha No.108 (ZC108), and Zijuan (ZJ). The study found a minor impact of cultivars on rhizosphere soil properties but a significant influence on microbial community structure. Except for available potassium (AK) (HJY > TGY > ZC108 > ZJ), tea cultivars had no significant impact on other soil properties. The tea cultivars resulted in substantial differences only in the diversity of soil bacteria of lower taxonomic levels (family to species), as well as significantly changed communities' structure of bacteria and fungi ($R^2$ = 0.184, $p$ = 0.013 and $R^2$ = 0.226, $p$ = 0.001). Specifically, Proteobacteria, Actinobacteria, Chloroflexi, Acidobacteriota, and Firmicutes accounted for approximately 96% of the bacterial phyla in the tea soils, while Ascomycota, Mortierellomycota, Rozellomycota, Basidiomycota, and Monoblepharomycota (90% of the total) predominated the soil fungal community. Redundancy analysis (RDA) identified soil pH (14.53%) and ammonium-nitrogen ($NH_4^+$-N; 16.74%) as the key factors for the changes in bacterial and fungal communities, respectively. Finally, FAPROTAX analysis predicted significant differences in the carbon, nitrogen, and sulfur (C-N-S)-cycling among the soils with different tea cultivars, specifically, ZJ cultivar showed the highest C-cycling but the lowest N- and S-cycling, while FUNGuild analysis revealed that the pathotroph group was significantly lower in ZC108 than the other cultivars. These findings improve our understanding of the differences in microbial community characteristics among tea cultivars and provide a basis for precisely selecting and introducing excellent tea varieties in the agriculture practices.

**Keywords:** rhizosphere microorganism; tea cultivars; microbial diversity; community composition; predicted function

## 1. Introduction

Soil microbes regulate ecological functions such as regulating soil nutrient cycle, decomposing organic matter, and inhibiting plant diseases [1]. These microbes are susceptible to environmental changes and anthropogenic perturbations, such as climate change, land use change, fertilization, irrigation, and crop replanting [2–4], and serve as key biological indicators of soil quality and health, soil microbes reciprocate rapidly in response to

changes in soil environment. Therefore, better understanding the response of soil microbial communities to agricultural management practices will help monitor and create optimal conditions for plant growth and productivity.

Soil microbes show high spatial variability and can be influenced by both biotic and abiotic factors. At a large spatial scale, meteorological variables and geographic location are the primary factors affecting soil microbial diversity and community composition, such as mean annual precipitation (MAP) and air temperature (MAT) [5]. Meanwhile, at a small spatial scale, soil pH and nutrient availability influence microbial properties [6,7]. Generally, neutral or slightly alkaline conditions favor vigorous growth of bacterial communities [8,9], while acidic environments support fungal growth [10]. Among the various nutrient components, soil organic matter (SOM), the primary source of carbon (C) and energy, plays an important role in the formation of microbial communities [11]. Studies have associated bacterial and fungal community structure with the quantity and quality of organic carbon in the soil [12].

Under natural conditions, SOM is mainly derived from plant litter, root residues, and their secretions, which depend highly on the vegetation type. Thus, crop cultivars influence the rhizosphere microenvironment and microbial community characteristics in an agricultural system. Zhang et al. (2021) [13] reported significant differences in the rhizosphere microbial communities' taxonomic, functional, and phenotypic composition across blueberry cultivars. This difference in microbial community composition is probably due to the differences in root exudates and litter in the soil that alter the rhizosphere environment, including soil pH and organic carbon content and composition [14]. Evidence suggests that significant difference in rhizosphere communities was observed between genotypes or cultivars within one species, due to root exudates released from cultivars or genotypes being able to recruit specific microbes to establish a unique community [15–17]. These earlier studies demonstrated the impact of species or cultivars of annual crops on soil microbial communities; however, perennial plants, such as the tea plant, remain poorly studied.

The tea plant (*Camellia sinensis* L.) is an important crop widely cultivated in tropical and subtropical regions of China. It is an aluminum (Al)-accumulating plant with high Al tolerance and prefers ammonium ($NH_4^+$) over nitrate ($NO_3^-$) [18]. Moreover, the soil under years of tea plant cultivation will become acidic, as confirmed by our long-term field experiment results that the natural acidification rate of tea plantation soil reached 0.071 units per year; this may be due to the organic acid in root exudates secreted by the roots of the tea plant [19,20]. Studies have also indicated that the concentration and composition of root exudates, which shape the microbial communities, are closely related to crop cultivars [15,17]. The tea plant, a perennial crop, is characterized by highly acidic soil conditions due to high N fertilizer supply and root activity. These characteristics probably create a specific rhizosphere micro-environment. However, earlier studies on tea plantation soil mainly focused on the changes in microbial communities under different fertilization treatments and planting years [3,21]. In addition, previously, bulk rather than rhizosphere soil was mostly focused on because of perennial woody of tea plants in which rhizosphere soil is relatively difficult to collect in comparison to bulk soil. For example, recently, Du et al. (2021) [22] demonstrated significant differences in the bulk soil fungal β-diversity, community composition, and functional guilds among four tea cultivars. It is generally accepted that the rhizosphere is the "hub" of interactions between plants and soil microbes [23]. Evidence showed that the rhizosphere microbes responded rapidly to the changes in environment, which could feed back plant health and nutrient deficiency [24]. Tea plant has regional adaptability and it needs to be maintained for many years once cultivated because of perennial crop. Thus, it is very important to select suitable varieties before cultivation, especially for a specific regional introduction. Consequently, better understanding the characteristics and changes in rhizosphere microbes in response to tea cultivars could help select and introduce tea cultivars.

The present study analyzed rhizosphere properties of four tea cultivars from a small block of tea plantation (started in the year of 2015). The specific objectives of this study are as follows: (1) Investigate the changes in soil properties, such as pH, available nutrients, and microbial community characteristics, including microbial diversity, community composition, and their potential functions across tea cultivars; (2) Identify the key soil factors driving the changes in rhizosphere microbial community. Our findings will advance our understanding of the effects of crop cultivars on the rhizosphere microenvironment and provide a basis for precisely selecting and introducing excellent tea varieties in the agriculture practices.

## 2. Materials and Methods

### 2.1. Site Description

The field experiment was carried out in Shengzhou (29.74° N, 120.82° E, 23 m above sea level) of the Tea Research Institute of the Chinese Academy of Agricultural Sciences. The experimental site had a northern subtropical humid climate, with a MAP of 1200 mm and MAT of 12.6 °C, which is suitable for tea cultivation. The soil of this region had an initial pH of 4.57 and organic carbon of 5.59 mg g$^{-1}$. Mixed fertilizers, including N fertilizer (urea, 150−450 pure N kg ha$^{-1}$), phosphate fertilizer (superphosphate, 90 P$_2$O$_5$ kg ha$^{-1}$), potassium fertilizer (potassium sulfate, 120 K$_2$O kg ha$^{-1}$), and organic fertilizer (rapeseed cake, 1500 kg ha$^{-1}$) were applied to the plantation to ensure the proper growth of tea plants. Split application of N fertilizer was carried out at the rate of 30%, 30%, 20% and 20% in February, May, July, and October based on the nutrient demand. Phosphorus and potassium fertilizer was applied as base fertilizer in October. All fertilizers were applied in furrows (10–15 cm deep) between two rows of tea trees and covered with soil. After the spring harvest, a heavy pruning was carried out at the end of April, and all pruned litter was left on the soil surface of the tea plantation.

### 2.2. Experimental Design

The tea plant cultivars were planted in 2015, in our study, four tea cultivars of Huangjinya (HJY), Tieguanyin (TGY), Zhongcha No.108 (ZC108), and Zijuan (ZJ) were chosen and were planted in a single row, with a plant spacing of 30 cm, a row spacing of 150 cm (Figure 1). Four tea cultivars were selected based on phenotype and growth activity. Each variety per one plot was planted in an area of approximately 36 m$^2$ (12 m × 3 m), with six replicates (plots). The planting density was about 20,000 plants per hectare.

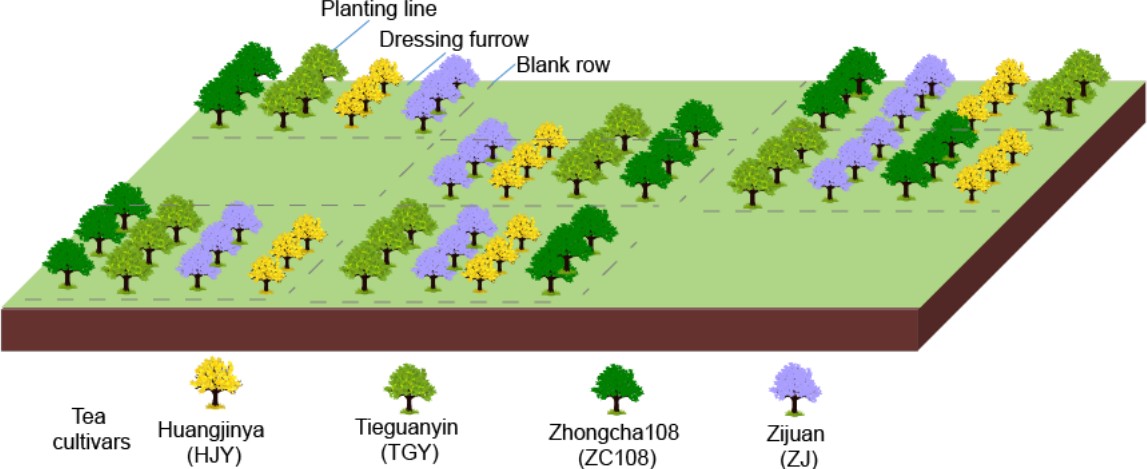

**Figure 1.** Schematic diagram of the experimental plot.

### 2.3. Soil Sampling

In April 2022, ten tea plants for each variety per plot were chosen, and approximately 5–6 live roots per each tea plant were obtained. The soil attached to the root surface was collected as the rhizosphere soil, and the collected samples were thoroughly mixed to obtain the composite sample. These mixed samples were immediately taken to the laboratory and split into three parts. One part was stored in a freezer at $-80$ °C for high-throughput sequencing, and the other part was stored in a refrigerator at 4 °C for ammonium-nitrogen ($NH_4^+$-N) and nitrate-nitrogen ($NO_3^-$-N) determination. The third part was further divided into two parts; one was air-dried and passed through a 2 mm sieve for pH and available nutrient analysis, and the other was ground and passed through a 0.15 mm sieve for total potassium (TK), and total phosphorus (TP) total nitrogen (TN), and soil organic carbon (SOC) analyses.

### 2.4. Soil Chemical Property Analysis, and Soil Fertility Index Calculation

The soil pH of a 1:2.5 suspension (soil: water; $w/v$) was measured using an Orion 3 Star pH meter (Thermo Ltd., Waltham, MA, USA). Meanwhile, $NH_4^+$-N and $NO_3^-$-N were extracted from 5 g of soil (dry weight) in 2 M KCl solution (1:10; w/w) and quantified using an ultraviolet spectrophotometer. SOC and TN were determined using an element analyzer (Vario Max, Elementar, Langenselbold, Germany). The available P (AP) and available K (AK) in the soil were extracted using Mehlich-3 solution [25] (soil: solution; 1:10 $w/v$), shaken at 180 rpm for 30 min at 25 °C, filtered through 0.45 μm filter membrane. The TK and TP in soil were digested in a microwave with a mixture of $HNO_3$: HCl: HF (5:2:2, $v:v:v$) (SINEO, MDS-6, Shanghai, China). The content of AP, AK, TP, and TK were measured by inductively coupled plasma-atomic emission spectrometry (ICAP6300, Thermo Fisher, USA). In our study, soil fertility index (SFI) was also constructed for comprehensively evaluating the soil nutrients based on the soil management assessment framework [26]. In brief, All the measured soil properties were included in a total data set (TDS). After establishing the TDS, the weight value of each soil parameter was calculated through a principal component analysis (PCA) [27]. Then, each parameter in the TDS was transformed and normalized to a value ranging between 0.1–1.0 using the standard scoring function (SSF) method. Three different SSF equations were selected to standardize the TDS parameters. Afterwards, the soil fertility index was determined as follows [28]:

$$\text{Soil fertility index } = \sum_{i=1}^{n} Wi \times Si$$

where $W$ is the weight value of each soil parameter, $S$ is the score of each parameter, and $n$ is the number of parameters in the TDS [26]. In general, a higher index score indicated a greater soil fertility.

### 2.5. DNA Extraction, PCR Amplification, and Amplicon Sequencing

Total DNA was extracted from 0.5 g of fresh soil using the Fast DNA® Spin Kit (MP Biomedical, Santa Ana, CA, USA) according to the manufacturer's instructions. The quality of the extracted DNA was assessed by agarose gel (1%) electrophoresis, and the concentration and purity of the DNA were measured using a Nanodrop® ND-2000 (Thermo Fisher Scientific Inc., Waltham, MA, USA). The bacterial 16S rRNA V5-V7 variable region was amplified using the 799F (5′-AACMGGATTAGATACCCKG-3′) and 1193R (3′-ACGTCATCCCCACCTTCC-5′) target primers [29], while the fungal ITS1 region was amplified with ITS1F (5′-CTTGGTCATTTAGAGGAAGTAA-3′) and ITS2R (5′-GCTGCGTTC TTCATCGATGC-3′) primers. Amplicons were purified using the Agarose Gel DNA Purification Kit (TaKaRa, Dalian, China). The purified amplicon of each sample was pooled and pair-end-sequenced (2 × 300 bp) on an Illumina MiSeq platform at Majorbio Bio-Pharm Technology Co. Ltd. (Shanghai, China) [30].

### 2.6. High-Throughput Sequencing Data Processing

The raw sequencing data were processed using the Quantitative Insights into Microbial Ecology (QIIME) software (Version 1.9.1) [31]. Low-quality sequences with lengths < 150 bp or with mononucleotide repeats were removed. The filtered pair-end reads were then assembled using the FLASH software (Version 1.2.11) [32]. Bacterial and fungal sequences in operational taxonomic units (OTUs) were clustered and compared to SILVA (Release 128, http://www.arb-silva.de, accessed on 22 December 2022) and NCBI (Release 6.0, http://unite.ut.ee/index.php, accessed on 22 December 2022) databases, respectively, using a similarity level of 97% [33,34]. Raw sequencing data have been deposited into the NCBI Sequence Read Archive (SRA, http://trace.ncbi.nlm.nih.gov/Traces/sra/sra.cgi (accessed on 22 December 2022)), with accession numbers PRJNA904044. Microbial richness and Shannon indices were calculated using the 'vegan' package in R (version 3.5.1).

### 2.7. Statistical Analysis

One-way analysis of variance (ANOVA) and Duncan's test were applied to evaluate the soil properties, fertility index, microbial numbers, alpha diversity, the relative abundance of dominant bacterial and fungal phyla, and their predicted functions among the different tea cultivars. Similarity of bacterial and fungal community structures was identified based on OTU abundance, using principal coordinate analysis (PCoA) based on Bray–Curtis distances, and corresponding differences among tea cultivars were tested by the permutation multivariate analysis of variance (PERMANOVA). Redundancy analysis (RDA) was performed to test the impact of soil properties on bacterial and fungal community structures, and PERMANOVA was used to identify the soil properties significantly affecting the bacterial and fungal community structures. Pearson correlation was used to examine the association among soil properties, microbial community characteristics, and their predicted functions. The linear discriminant analysis (LDA) effect size (LEfSe) analysis was used to detect the potential biomarkers at multiple taxonomical levels with the strongest effect, using an alpha value of 0.05 for the factorial Kruskal–Wallis test and an LDA score threshold of 3.0. Further, the functional profiles of bacterial and fungal communities were annotated using FAPROTAX and FUNGuild databases on the Majorbio Cloud Platform [30].

All statistical analyses were performed using the R platform (version 4.1.2). The 'stats' package was used for Pearson correlation analysis and one-way ANOVA, and the 'vegan' package was used for RDA, PCoA, and PERMANOVA.

## 3. Results

### 3.1. Changes in Soil Properties and Fertility Index

The present study found no significant differences in the soil chemical properties, except for AK, among the four tea cultivars (Table 1). The analysis revealed the highest AK content in HJY, followed by TGY, ZC108, and ZJ. Meanwhile, the highest fertility index was observed in TGY and the lowest in ZC 108; however, the differences among the cultivars were not significant.

**Table 1.** Soil chemical properties under different tea cultivars.

| Parameters | Tea Cultivars | | | |
| --- | --- | --- | --- | --- |
| | HJY | TGY | ZC108 | ZJ |
| pH ($H_2O$) | 3.95 (0.10) [a] | 3.94 (0.06) [a] | 3.85 (0.07) [a] | 3.81 (0.07) [a] |
| $NH_4^+$ (mg kg$^{-1}$) | 1.30 (0.29) [a] | 1.71 (0.31) [a] | 1.46 (0.25) [a] | 1.76 (0.45) [a] |
| $NO_3^-$ (mg kg$^{-1}$) | 4.42 (0.26) [a] | 4.64 (0.65) [a] | 4.47 (0.34) [a] | 4.24 (0.21) [a] |
| AP (mg kg$^{-1}$) | 1.85 (0.54) [a] | 1.57 (0.34) [a] | 0.87 (0.18) [a] | 1.23 (0.30) [a] |
| AK (mg kg$^{-1}$) | **286 (19)** [a] | **252 (22)** [ab] | **217 (14)** [bc] | **197 (13)** [c] |
| TP (mg kg$^{-1}$) | 424 (30) [a] | 471 (28) [a] | 399 (36) [a] | 452 (38) [a] |
| TK (mg kg$^{-1}$) | 5527 (498) [a] | 4983 (680) [a] | 5020 (704) [a] | 5316 (469) [a] |

**Table 1.** *Cont.*

| Parameters | Tea Cultivars | | | |
|---|---|---|---|---|
| | HJY | TGY | ZC108 | ZJ |
| SOC (g kg$^{-1}$) | 9.1 (0.5) [a] | 9.3 (0.5) [a] | 9.4 (0.5) [a] | 8.5 (0.4) [a] |
| TN (g kg$^{-1}$) | 1.2 (0.1) [a] | 1.2 (0.1) [a] | 1.2 (0.0) [a] | 1.1 (0.0) [a] |
| C/N | 7.60 (0.24) [a] | 7.83 (0.16) [a] | 7.82 (0.20) [a] | 7.48 (0.20) [a] |
| SFI | 0.55 (0.07) [a] | 0.58 (0.04) [a] | 0.52 (0.05) [a] | 0.55 (0.07) [a] |

Data are shown as mean (standard error). Different letters in the same column indicate significant differences among the cultivars at $p < 0.05$. Significant differences ($p < 0.05$) among tea cultivars are shown in bold. SFI, soil fertility index, $NH_4^+$, ammonium, $NO_3^-$, nitrate, AP, available phosphorus, AK, available potassium, TP, total phosphorus, TK, total potassium, SOC, soil organic carbon, TN, total nitrogen. HJY, Huangjinya, TGY, Tieguanyin, ZC108, Zhongcha108, ZJ, Zijuan.

### 3.2. Rhizosphere Microbial Richness and Diversity

The study found differences in bacterial and fungal OTUs among the soils with different tea cultivars. A total of 1687 bacterial OTUs were identified in the soil with TGY, 1704 in HJY, 1594 in ZJ, and 1673 in ZC108 (Figure 2), including 88 unique OTUs of TGY, 89 of HJY, 71 of ZJ, and 92 of ZC108. The four tea cultivars shared 1171 OTUs, which accounted for 54.06% of the total. Meanwhile, 1673 fungal OTUs were found in the soil with TGY, 1600 in HJY, 1556 in ZJ, and 1665 in ZC108, including 281 unique OTUs of TGY, 261 of HJY, 262 of ZJ, and 269 of ZC108. A total of 739 OTUs were shared among the four cultivars.

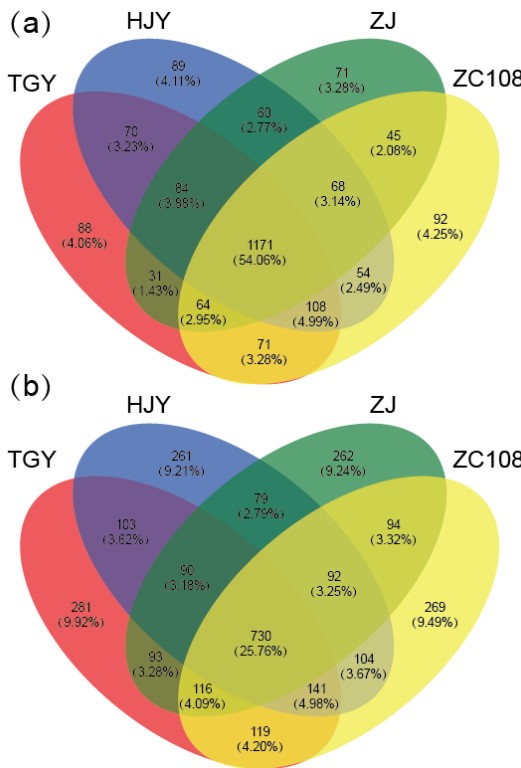

**Figure 2.** Venn diagram of bacterial (**a**) and fungal (**b**) OTUs in the soils with different tea cultivars. HJY, Huangjinya, TGY, Tieguanyin, ZC108, Zhongcha108, ZJ, Zijuan.

Further, the microbial population from phylum to species levels that determined the richness and diversity based on Chao1 and Shannon indices were analyzed, respectively (Table 2). Overall, only bacterial populations were found to be sensitive to the tea cultivars. Significant differences were observed in the soil bacteria at the lower taxonomic levels (family to species) among the tea cultivars, while soil bacteria at the higher taxonomic levels (phylum to order) showed no significant difference. The highest population was found in

HJY and the lowest population in ZJ. A similar trend was observed in the Shannon index in which HJY showed the higher Shannon index value and ZJ showed the lowest, and there was no significant difference in the Chao1 index; however, no significant difference was observed in the population and diversity (Chao1 and Shannon indices) of soil fungi among the four tea cultivars.

The PCoA showed significant differences in soil bacterial and fungal communities among the four tea cultivars (PERMANOVA: $r^2 = 0.184$, $p < 0.05$ for the bacterial community; $r^2 = 0.226$, $p < 0.001$ for the fungal community; Figure 3). Here, the two axes (PCoA1 and PCoA2) together explained 35.87% and 33.19% of the total variation in the bacterial and fungal communities, respectively.

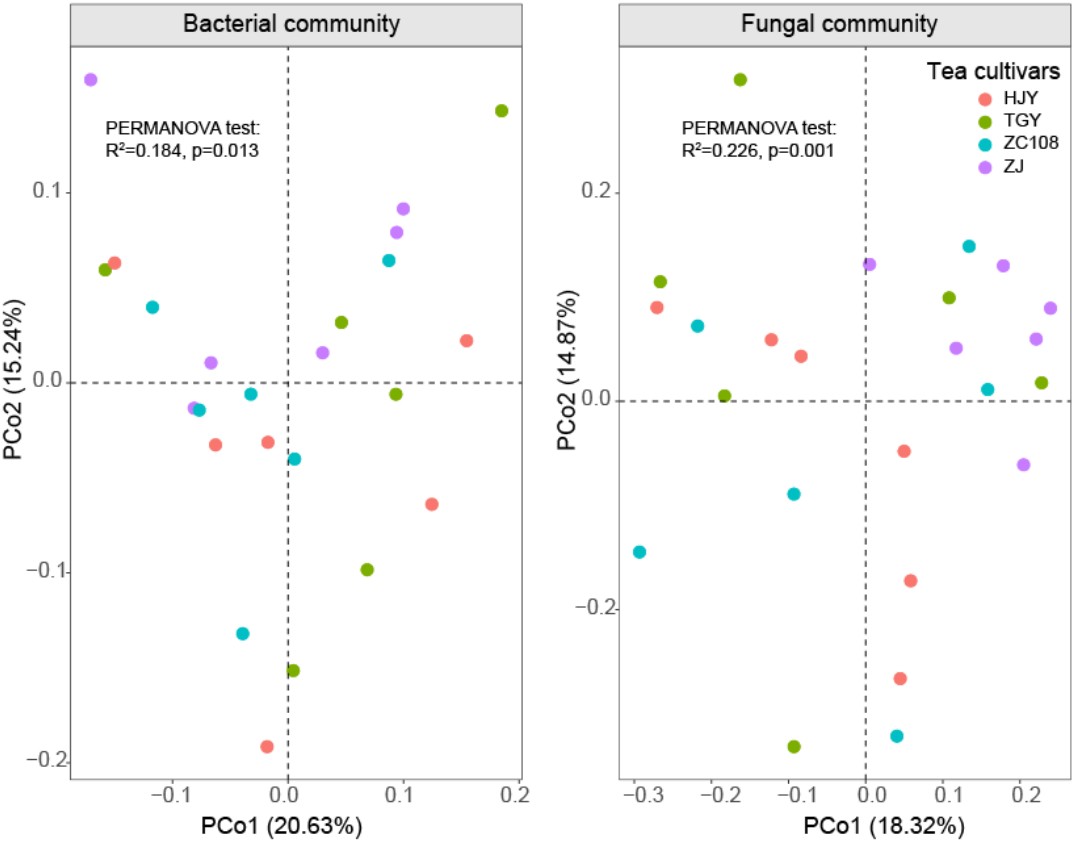

**Figure 3.** Principal coordinate analysis (PCoA) of bacterial (**left** panel) and fungal (**right** panel) community composition at the OTU level based on Bray–Curtis distances. The percentage variations explained by PCoA1 and PCoA2 are shown along the axes. A permutational multivariate analysis of variance (PERMANOVA) was used to determine the significant effect of the tea cultivars.

**Table 2.** The number, richness, and diversity of bacterial and fungal OTUs (97% similarity level) in soils with four different tea cultivars.

| Parameters | Tea Cultivars | | | |
|---|---|---|---|---|
| | HJY | TGY | ZC108 | ZJ |
| **Bacteria** | | | | |
| Phylum | 21 (0.4) [a] | 21 (0.5) [a] | 20 (0.4) [a] | 20 (0.4) [a] |
| Class | 42 (0.8) [a] | 42 (1.3) [a] | 40 (1.5) [a] | 40 (1.8) [a] |
| Order | 101 (1.4) [a] | 101 (2.0) [a] | 99 (3.3) [a] | 96 (1.9) [a] |
| Family | **159 (1.6) [a]** | **158 (3.5) [a]** | **155 (3.7) [ab]** | **149 (3.0) [b]** |
| Genus | **271 (5.1) [a]** | **266 (6.8) [ab]** | **439 (5.0) [ab]** | **422 (5.8) [b]** |
| Species | **452 (8.8)[a]** | **445 (10.6) [ab]** | **439 (9.5) [ab]** | **422 (9.0) [b]** |

**Table 2.** *Cont.*

| Parameters | Tea Cultivars | | | |
|---|---|---|---|---|
| | HJY | TGY | ZC108 | ZJ |
| Chao1 | 1267 (26) [a] | 1267 (21) [a] | 1253 (38) [a] | 1202 (12) [a] |
| Shannon | **5.11 (0.02) [a]** | **4.96 (0.07) [ab]** | **5.05 (0.06) [ab]** | **4.92 (0.05) [b]** |
| **Fungi** | | | | |
| Phylum | 12 (0.2) [a] | 12 (0.2) [a] | 12 (0.2) [a] | 12 (0.2) [a] |
| Class | 33 (0.7) [a] | 34 (0.3) [a] | 33 (0.7) [a] | 32 (1.0) [a] |
| Order | 73 (2.1) [a] | 75 (1.7) [a] | 72 (2.4) [a] | 71 (2.9) [a] |
| Family | 147 (4.2) [a] | 151 (4.4) [a] | 141 (4.3) [a] | 143 (2.9) [a] |
| Genus | 221 (8.1) [a] | 226 (6.7) [a] | 216 (7.7) [a] | 210 (11.2) [a] |
| Species | 283 (9.9) [a] | 285 (6.0) [a] | 273 (8.9) [a] | 367 (14.0) [a] |
| Chao1 | 691 (35) [a] | 734 (30) [a] | 700 (30) [a] | 637 (35) [a] |
| Shannon | 3.71 (0.14) [a] | 3.63 (0.13) [a] | 3.64 (0.09) [a] | 3.86 (0.10) [a] |

Data are presented as mean (standard error). Different alphabets in the same row indicate significant differences at $p < 0.05$ among the tea cultivars. Significant differences ($p < 0.05$) among tea cultivars are shown in bold. HJY, Huangjinya, TGY, Tieguanyin, ZC108, Zhongcha108, ZJ, Zijuan.

### 3.3. Rhizosphere Microbial Community Composition

Detailed analysis revealed Proteobacteria (34.51%), Actinobacteriota (26.33%), Chloroflexi (21.19%), Acidobacteria (11.97%), and Firmicutes (2.07%) that accounted for more than 96% of the bacterial community were the predominant phyla in the tea plantation soils (Figure 4). However, no significant difference was observed in the top 10 bacterial phyla, except for Armatimonadota, among the different soils (Figure S1).

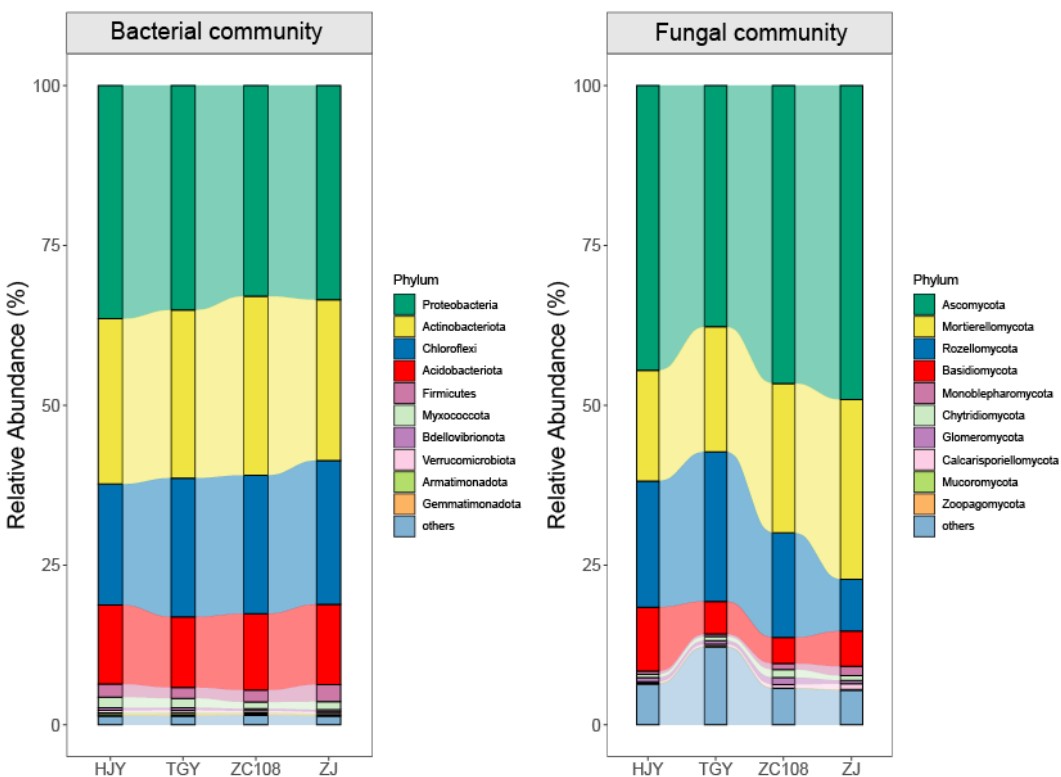

**Figure 4.** The relative abundance of the top 10 bacterial (**left** panel) and fungal (**right** panel) taxa at the phylum levels in the tea plantation soils under four tea plant cultivars. HJY, Huangjinya, TGY, Tieguanyin, ZC108, Zhongcha108, ZJ, Zijuan.

Meanwhile, the predominant Ascomycota (44.5%), Mortierellomycota (22.1%), and Rozellomycota (16.9%) together accounted for nearly 83.5% of the fungal community

(Figure 4). Among the top 10 fungal phyla, only Monoblepharomycota, Chytridiomycota, Mucoromycota, and Zoopagomycota were significantly different among the four tea cultivars (Figure S1); however, these phyla were less abundant in the soils. In addition, Pearson correlation revealed that the relationship between bacterial and fungi taxa at the phylum level was dominated by negative correlation (Table S1).

### 3.4. Microbial Community of Soils with Different Tea Cultivars

Then, LEfSe analysis was performed to identify the biomarker taxa at the phylum to genus levels with significantly different abundances among the four cultivars (Figure 5). The study identified 25 bacterial taxa and 56 fungal taxa differentially abundant in the soils (Figures S2 and S3). In the bacterial community, the Alphaproteobacteria class was identified as the most abundant biomarker ($p < 0.01$) in HJY, while Arthrobacter and Paenarthrobacter were the most differentially abundant genera in TGY. Meanwhile, Pseudomonadaceae and Xanthobacteraceae families were found enriched in the soils with ZJ and ZC108. From the fungal community, more than ten biomarker taxa were identified in HJY (14), TGY (15), and ZJ (21), while only six were enriched in ZC108.

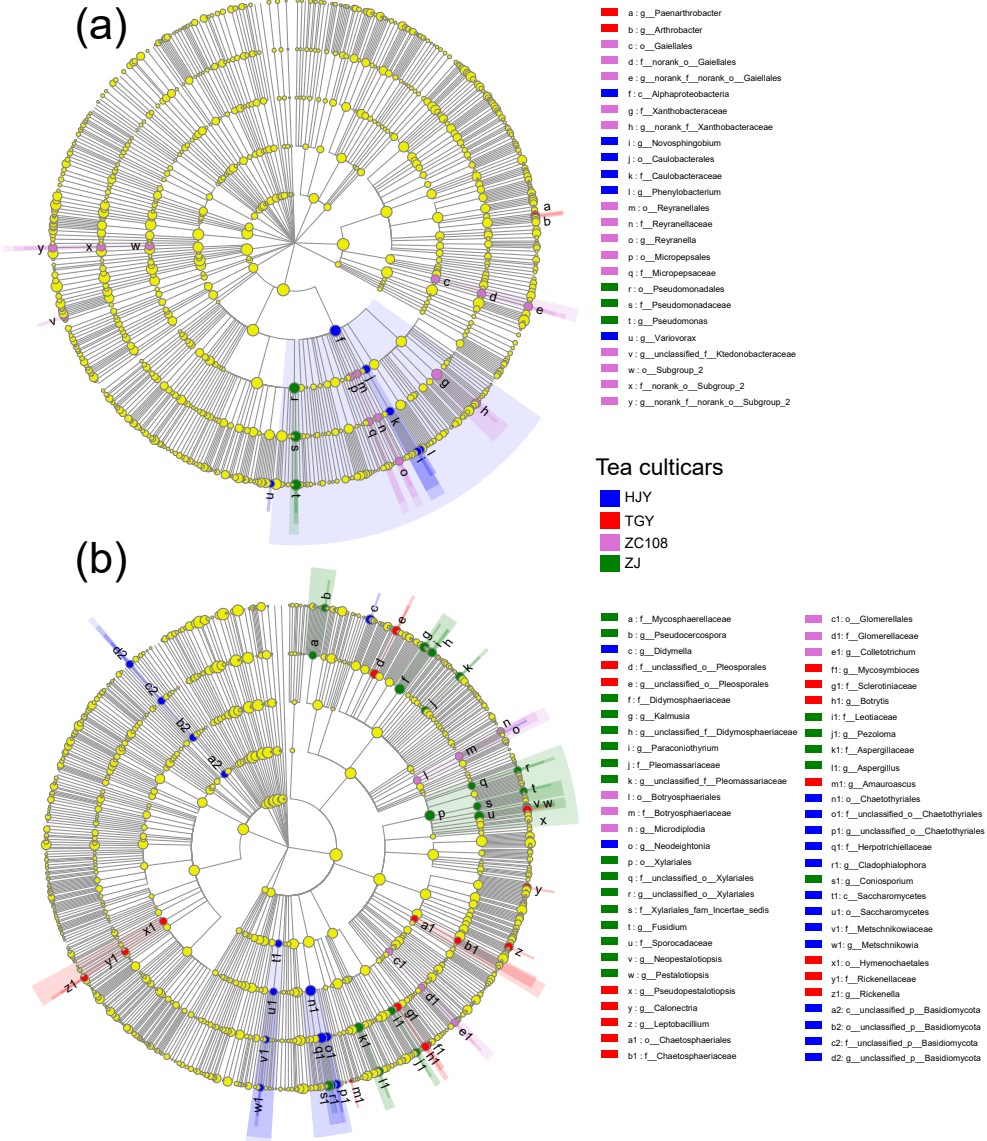

**Figure 5.** Linear discriminant analysis (LDA) effect size (LEfSe) taxonomic cladogram highlights the bacterial (**a**) and fungal (**b**) biomarkers in soils with four different tea plant cultivars. Four tea cultivars

included HJY, Huangjinya, TGY, Tieguanyin, ZC108, Zhongcha108, and ZJ, Zijuan. Significantly discriminant taxon nodes are colored and branch areas are shaded according to the highest ranked variety for that taxon. Non-significant discriminant taxonomic nodes are colored yellow. The five rings of the cladogram stand for phylum (innermost), class, order, family, and genus (outermost). Taxa with an LDA score > 3.0 were ultimately identified as the biomarkers.

### 3.5. Correlation between Microbial Communities and Soil Properties

We then performed RDA to assess the correlation between microbial communities and soil properties at the OTU level (Figure 6). For the bacterial community, the first and second dimensions (RDA1 and RDA2) of the ordination space from the RDA explained 15.37% and 10.39% of the total variances in the bacterial community. Then, the PERMANOVA test identified pH, TP, $NH_4^+$, AP, and SOC as key factors driving bacterial community; pH explained 14.53% variation, followed by $NH_4^+$ (7.57%), TP (7.29%), AP (6.11%), and SOC (6.08%). In addition, SOC and TN positively correlated with bacterial α-diversity (Shannon and Chao1) ($p < 0.05$; Table S2). Meanwhile, AK showed a significant positive correlation with the low-abundant taxa at the phylum level, such as Myxococcota (r = 0.57, $p < 0.01$), Bdellovibrionota (r = 0.46, $p < 0.05$), Verrucomicrobiota (r = 0.55, $p < 0.01$), and Armatimonadota (r = 0.47, $p < 0.05$). For the fungal community, RDA1 and RDA2 explained 11.23% and 9.63% of the total variances. PERMANOVA test identified $NH_4^+$ (16.74%) and TN (6.75%) as the significant variable driving changes in fungal community structure. The Chao1 positively correlated with pH (r = 0.50, $p < 0.05$) and AK (r = 0.56, $p < 0.01$), while the Shannon index negatively correlated with SOC (r = −0.47, $p < 0.05$) and TN (r = −0.48, $p < 0.05$). Meanwhile, pH and AK negatively correlated with Monoblepharomycota (r = −0.67 and −0.54, $p < 0.01$) and Chytridiomycota (r = −0.59, $p < 0.01$ and r = −0.41, $p < 0.05$) (Table S3).

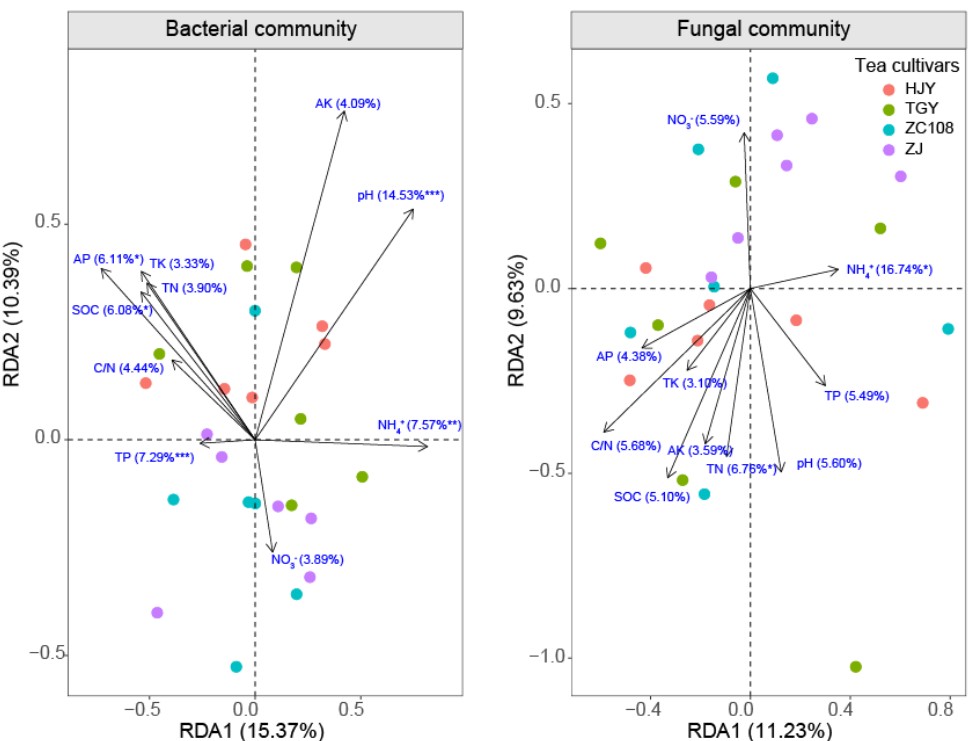

**Figure 6.** Redundancy analysis (RDA) shows the relationship between bacterial (**left** panel) and fungal (**right** panel) community composition and soil properties. The values in brackets represent the explained variation of each soil property in bacterial and fungal community composition. The significance level is indicated by * ($p < 0.05$), ** ($p < 0.01$) and *** ($p < 0.001$) by using a permutational multivariate analysis of variance (PERMANOVA). HJY, Huangjinya, TGY, Tieguanyin, ZC108, Zhongcha108, ZJ, Zijuan.

### 3.6. Potential Functions of Microbial Community

Finally, we used FAPROTAX to predict the biochemical cycle of environmental samples. The results showed significant differences in the relative abundance of soil microbial communities associated with C, N, and S cycles under different tea cultivars (Figure 7). The C-cycling microbes were the highest in ZJ and the lowest in TGY. On the contrary, the microbes associated with N and S cycles were the most abundant in TGY soil and the least in ZJ soil. Further Pearson's analysis revealed that C-cycling abundance negatively correlated with soil pH (r = −0.66, $p < 0.01$) and AK (r = −0.76, $p < 0.01$) (Table S4), while N-cycling abundance positively correlated with soil pH (r = 0.51, $p < 0.05$) and AK (r = 0.59, $p < 0.01$); S-cycling abundance positively correlated with AK (r = 0.41, $p < 0.05$) but negatively with C/N (r = −0.44, $p < 0.05$). Most bacterial phyla significantly correlated with C-cycling in which a positive correlation was observed for Actinobacteriota and Firmicutes and a negative correlation for Myxococcota, Bdellovibrionota, Verrucomicrobiota, and Armatimonadota. Among the fungal phyla, only Monoblepharomycota positively correlated with C-cycling. Meanwhile, N-cycling negatively correlated with the bacterial phylum Firmicutes and the fungal phylum Monoblepharomycota and positively with the bacterial phylum Verrucomicrobiota. On the other hand, S-cycling negatively correlated with the bacterial phylum Acidobacteriota alone; however, no significant correlation was observed between S-cycling and fungal phyla.

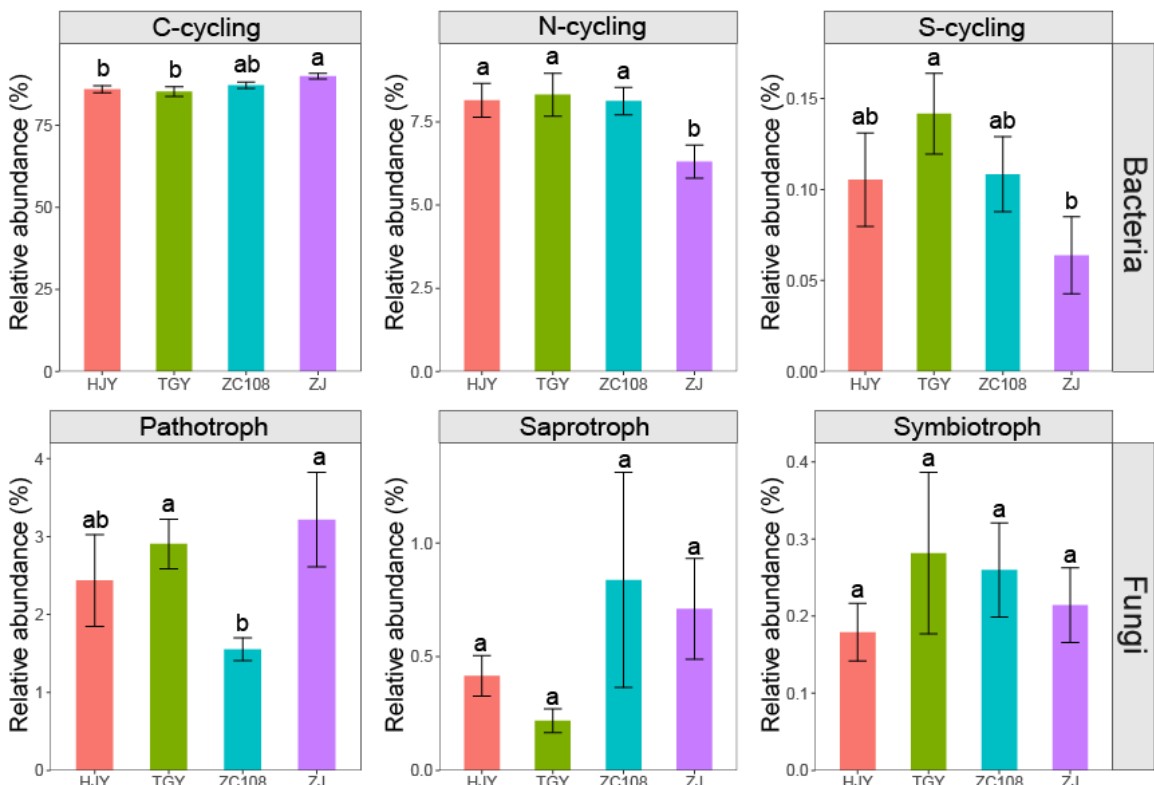

**Figure 7.** The major functional profiles of bacterial (**top** panel; C, N, and S cycle by FAPROTAX) and fungal (**bottom** panel; Pathotroph, Saprotroph, and Symbiotroph by FUNGuild) communities. The data shown are mean ± standard error (SE; n = 6). Different lowercase letters above the bars indicate significant differences ($p < 0.05$). HJY, Huangjinya, TGY, Tieguanyin, ZC108, Zhongcha108, ZJ, Zijuan.

Further, FUNGuild was used to predict the functional groups of the fungal communities under tea cultivars. The FUNGuild database divided the fungi of these soils into saprotroph, pathotroph, and symbiotroph categories based on their nutrition mode (Figure 7). Here, pathotroph was the dominant functional group (2.35% of the total), followed by saprotroph (0.55%) and symbiotroph (0.23%). A significant difference ($p < 0.05$)

was observed in the saprotroph group among the tea cultivars; the relative abundance of saprotroph in the soil with ZC108 was significantly lower than that in the soils with TGY and ZJ. Moreover, only pathotroph negatively correlated with SOC (r = −0.52, $p < 0.05$), TN (r = −0.47, $p < 0.05$), and C/N (r = −0.43, $p < 0.05$). Meanwhile, no significant correlation was observed between the saprotroph or symbiotroph and the soil properties (Table S4). A weak correlation was found between the fungal functional groups and the top 10 bacterial and fungal phyla. The saprotroph negatively correlated with the bacterial phylum Armatimonadota (r = −0.45, $p < 0.05$) and the fungal phylum Rozellomycota (r = −0.44, $p < 0.05$) but positively with the fungal phylum Ascomycota (r = 0.62, $p < 0.01$). Meanwhile, symbiotroph was positively correlated with the bacterial phylum Gemmatimonadota alone (r = 0.44, $p < 0.05$) (Table S5).

## 4. Discussion

### 4.1. Changes in Soil Properties among Different Tea Cultivars

Studies have confirmed the significant impact of crop cultivars on the rhizosphere soil environment. For example, Jiang et al. (2016) [35] found that soil properties, such as SOC, TN, and TP, were substantially different among the blueberry cultivars. Meanwhile, Kong et al. (2020) [36] reported differences in rhizosphere soil properties among maize cultivars. Evidence shows that root exudates affect the soil microenvironment [23]. Root exudates are composed of different forms of organic carbon that directly affect the quantity and quality of C. Then, soil microorganisms use these root exudates as a C source and influence soil processes, such as nitrification [37]. Moreover, the organic acids in the root exudates, such as oxalic, malic, and fumaric acid, have a substantial impact on soil pH by releasing proton ($H^+$) directly. Root exudates also contain various enzymes, including those of C, N, and phosphorus (P) cycles, these enzymes participated in and catalyzed a series of complex biochemical reactions in soil. Researchers have indicated that the root exudates vary greatly based on genetic characteristics, indicating that plant genotype or crop variety of one species exert a substantial impact on various soil enzyme activities, thus explaining the differences in soil properties under different varieties [38,39].

Recently, Du et al. (2022) [22] reported a significant difference in the soil properties, such as pH and SOC, among the tea cultivars Chuancha No. 3, Chuanmu No. 217, Chuannong Huangyazao, and *C. sinensis* 'Fuding Dabaicha'. However, in our study, tea cultivars showed a minor effect on rhizosphere soil properties (Table 1), probably due to soil heterogeneity and the short experimental duration. In our study, experimental tea plants were distributed across three big blocks (upper, middle, and lower position) (Figure 1). Moreover, experimental duration was considered as an important factor affecting results, which had been confirmed by most previous studies [40]. Therefore, the long-term monitoring is particularly important in clarifying the experimental results, especially in the fields. In addition, it is generally accepted that soil chemical properties responded poorly to environmental stress in comparison to microbial properties [41]; therefore, this may be the reason why the rhizosphere soil properties have not changed significantly with the seven-year tea plant cultivation. Meanwhile, a significant difference was found in available potassium (AK) among the soils with different tea cultivars in which the highest was in HJY, and the lowest was in ZJ (Table 1). This difference might be due to the difference in the K absorbed by the cultivars. It is generally believed that ZJ is a variety with large leaves and has high biomass; therefore, it absorbs more K from the soil and reduces residual K. Meanwhile, HJY has small leaves and weak growth potential; it absorbs less K, resulting in relatively higher residual K in the soil; reminding that P and K nutrients should also be considered in the filed management according to nutrient demand of different varieties. This finding implies that different tea varieties need corresponding field management strategies. Additionally, soil properties may not be the best indicator to feedback crop stress in some cases.

### 4.2. Culitvar-Associated Response Characteristics of Soil Microbial Diversity, Community Composition, and Their Potential Functions

The rhizosphere microbes are key determinants promoting plant growth and productivity; they participate in nutrient absorption, regulate plant metabolism, and activate biotic and abiotic stress responses [42]. The plant genotype or crop cultivar affects the root-related microbiome. Several studies have demonstrated significant differences in the rhizosphere microbial community among plant varieties [43] or genotypes [44]. In our study, a significant difference was observed in the diversity and population of rhizosphere bacteria among tea cultivars; however, no significant difference was observed in the rhizosphere fungi (Table 2), probably because the bacteria are more sensitive to environmental changes and anthropogenic perturbations than the fungi [45]. Moreover, bacteria have a much shorter turnover time than fungi and are more sensitive to the availability of C substrates in the soil [46,47]. Consequently, rhizosphere bacteria responded rapidly to the different tea cultivars (Table 2). This finding suggests that the soil bacteria may be a sensitive indicator for evaluating the rhizosphere effect in response to the changes in biotic and abiotic stresses, especially in a short time.

In our study, the bacterial community of the tea rhizosphere soil was dominated by Proteobacteria, Actinobacteriota, Chloroflexi, Acidobacteria, and Firmicutes (95%) (Figure 4), consistent with Li et al. (2016) [21]. Meanwhile, the fungal community was dominated by Ascomycota, Mortierellomycota, and Rozellomycota phyla. The abundant phyla bacteria and fungi were consistent with those reported earlier, indicating them as the generalist taxa that fulfill general ecosystem functions [48]. Meanwhile, many soil ecosystem functions were also performed by the low-abundant taxa rather than dominant taxa [49]. We specifically found significant differences in the microbial population at the lower taxonomic levels (family or species) among the four tea cultivars but no difference at the higher taxonomic levels (phylum or class) (Table 2). Studies have demonstrated that certain environmental conditions select specific microbial taxa of lower taxonomic levels [50,51], and this regulation of low-abundant microbial taxa helps improve soil fertility [52]. Thus, the present study's findings and the earlier reports suggest the use of lower microbial taxa as potential indicators of the soil trophic status, plant health, and agro-ecosystem stability [53] in plantations under different tea cultivars.

In this study, tea cultivars only had a significant impact on soil bacterial diversity (Table 2), but bacterial and fungal community structures are significantly different among different tea cultivars (Figure 3). This is mainly due to changes in the rhizosphere microenvironment. Based on RDA modeling, soil pH was identified as the most important variable affecting the rhizosphere bacterial community (Figure 6), consistent with the earlier reports [54]. Studies have confirmed that microbes grow more vigorously under neutral or slightly alkaline conditions [9] and exhibit poor growth in low-pH environments [8]. This difference in growth is likely because a strongly acidic environment inhibits microbial enzyme activity and cellular metabolism [55]. In agreement with this, our study found a positive correlation between the Chao1 index of bacteria and soil pH (Table S2). Although soil pH showed no significant difference among tea cultivars, an obvious change trend was observed, indicating that small changes in soil pH can cause significant changes in soil microbial community structure [56]. Moreover, $NH_4^+$ was identified as the key factor driving the change in fungal community structure (Figure 6), probably associated with the growth of plants. Under normal conditions, soil N availability largely contributes to plant growth, and its absorption by plant is highly dependent on rhizosphere microbial guilds [57]. The tea plant is a perennial crop with high N demand and prefers $NH_4^+$ over $NO_3^-$ [18], and $NH_4^+$ absorption capacity varies largely among tea cultivars [58]; in our study, although no significant difference of soil $NH_4^+$ was observed among cultivars, similar to the pH change trend, tiny changes in soil $NH_4^+$ exerted a significant impact on fungal community structures of rhizosphere soils (Figure 6), in agreement with Pajares et al.'s (2018) [59] result that small changes in N availability in the black soils can cause large changes in microbial

community structure. This finding implies that nitrogen supply should be depended on crop cultivars in practices.

Further, FAPROTAX showed significant differences in element cycling between the ZJ rhizosphere soil and the other three soils with tea cultivars (Figure 7). Specifically, the relative abundance of C-cycling microbes in the soil with ZJ was significantly higher than the others. In contrast, the relative abundance of N- and S-cycling microbes in ZJ was considerably lower, probably due to the difference in root exudates. The ZJ cultivar contains high anthocyanin content, which substantially affects microbiota [60]. Therefore, we suspect that the composition of root exudates is an essential factor influencing soil microbial activity, consequently, causing a substantial impact on soil functions [15]. In our study, N-cycling function responded rapidly to cultivars (Figure 7), indicating that soil microbes play a key role in soil nutrient transformation and absorption. Meanwhile, pathotroph fungi accounted for a significant portion, while the symbiotroph accounted for only a small proportion among three groups (Figure 7), probably due to the growth of a series of pathogenic microbes under perennial planting conditions. Thus, the adverse impact warns us to pay attention to soil maintenance under continuous tea cultivation. In our study, it should be noted that the pathotroph group was significantly lower in ZC108 than in the other cultivars, in agreement with ZC108 that has good stress resistance, implying that ZC108 may be an optional variety for continuous cropping tea plantation.

## 5. Conclusions

The present study observed a minor impact of tea cultivars on soil properties but a significant influence on rhizosphere microbial properties. We found differences in the rhizosphere microbial community structure among tea cultivars and identified soil pH and $NH_4^+$ as the key factors driving bacterial and fungal communities. The ZJ cultivar promoted the rhizosphere soil's C cycle but lowered N and S cycles. In addition, pathotroph fungi accounted for the largest proportion of the fungal functional guilds under all cultivars in which ZC108 cultivar showed the lowest pathotroph fungi abundance, indicating that continuous cropping obstacles also exist under tea plant cultivation, even in the short term, and ZC108 cultivar may have strong resistance. These findings suggest that some specific microbial strains linked to nutrient cycles should be selected and used as "starter" cultures to promote and accelerate growth of new tea plantations.

**Supplementary Materials:** The following are available online at https://www.mdpi.com/article/10.3390/agronomy13010221/s1. Figure S1: bubble plots show the relative abundance of the top 10 bacterial (left panel) and fungal composition (right panel) at the phylum levels in the tea plantation soils under different tea cultivars. The size of the bubble corresponds to the relative abundance of microbial taxa. Different lowercase letters on the right of the bubbles indicate significant differences among the treatments (one-way ANOVA test, $p < 0.05$). Figure S2: histogram of the LDA scores computed for differentially abundant soil bacteria under different tea cultivars' planting. Figure S3: histogram of the LDA scores computed for differentially abundant soil fungi under different tea cultivars' planting. Table S1: the correlations of bacterial and fungal community composition at the phylum levels. Table S2: the correlations of Chao1, Shannon index, and composition (phylum level) of soil bacterial community with soil properties. Table S3: the correlations of Chao1, Shannon index, and composition (phylum level) of soil fungal community with soil properties. Table S4: the correlations of bacterial predicted function and fungal predicted function with soil properties. Table S5: the correlations of bacterial predicted function and fungal predicted function with community composition of soil bacteria and fungi.

**Author Contributions:** S.W.: Investigation, Data Curation, Writing—Original Draft; B.L.: Resources, Data curation, Visualization; K.N.: Methodology, Software; L.M.: Investigation; Y.S.: Investigation; Y.L.: Investigation; S.Z.: Software; S.G.: Supervision, Conceptualization; X.Y.: Writing—Review and Editing, Funding acquisition; J.R.: Supervision, Funding Acquisition, Conceptualization. All authors have read and agreed to the published version of the manuscript.

**Funding:** This research was supported by the Yunnan provincial special fund for the construction of a science and technology innovation base (202102AE090038), the China Agriculture Research System

of MOF and MARA (CARS-19), the Central Public-interest Scientific Institution Basal Research Fund (Y2022LM26), Special Funds for Basic Research and Operation of Tea Research Institute, Chinese Academy of Agricultural Sciences (1610212022008), and Science and Technology Innovation Fund Project of Fujian Agriculture and Forestry University (KFb22109XA).

**Institutional Review Board Statement:** Not applicable.

**Informed Consent Statement:** Not applicable.

**Data Availability Statement:** Not applicable.

**Conflicts of Interest:** The authors declare no conflict of interest.

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
