# Peer review of "Rhizosphere Microbial Community Shows a Greater Response Than Soil Properties to Tea (Camellia sinensis L.) Cultivars"

_agronomy, doi:10.3390/agronomy13010221_

Round 1

Reviewer 1 Report

The current study is on a topic of relevance and general interest to the readers and journals. This article talks about the observation of the tea cultivar impact on soil fertility and microbial community composition. This topic has great interest to the community, and it is in line with the aims of the journal (in the special issue). Although the manuscript is written with a proper English, I found that the authors have to improve few parts. However, the manuscript is clear and well written, it could be easily managed and be ready for publication.

Rows 3-4: The use of the terms “under 3 seven-year cultivation” could be misunderstood, and readers could interpret this time unit as if you collected data for 7 years and not the seventh year. Therefore, I suggest removing this information from the title.

Row 17: “transformation” in my opinion the term transformation without some examples is reductive. It is more effectiveness the term cycle, because this term is more understandable and includes nutrient transformation and plant-soil-nutrient enrollment.

Rows 86-87: if you write “studies” you must cite them or show them close to your affirmation. If you referred to Du et al. 2021 it is only one study. Please check if there are more citation about this sentence.

Rows 227-229: It is better to avoid the direct speech in scientific papers, because it looks not professional. Therefore, it will be better to change this sentence into a passive form.

Rows 369-372: when you write that “researchers” do something you must put their references, because you are affirming that someone else has done something, and this need to be supported.

Rows376-379: In my opinion, since you have observed an effect on microbial communities by the different cultivars, it would be desirable that the selection of microbial strains (specific of the tea cultivars), which are linked to nutrient cycles, can be selected and used as "starter" cultures to promote and accelerate growth of new tea plantations Please add some considerations about this effect, both here in the conclusion and in the abstract.

Author Response

Point-by-point responses to the reviewers' comments

To Reviewer #1:

The current study is on a topic of relevance and general interest to the readers and journals. This article talks about the observation of the tea cultivar impact on soil fertility and microbial community composition. This topic has great interest to the community, and it is in line with the aims of the journal (in the special issue). Although the manuscript is written with a proper English, I found that the authors have to improve few parts. However, the manuscript is clear and well written, it could be easily managed and be ready for publication.

Author Response: Thank you for your comments and suggestions. These comments and suggestions are very helpful for improving the quality of our study. In our revised version, parts of them were carefully revised for improving the readability according to reviewer’s suggestions.

Rows 3-4: The use of the terms “under 3 seven-year cultivation” could be misunderstood, and readers could interpret this time unit as if you collected data for 7 years and not the seventh year. Therefore, I suggest removing this information from the title.

Author Response: Thank you for kind reminding. In the revised manuscript, the terms of “under seven-year cultivation” was removed from the original manuscript title. Please see line 2-3.

Row 17: “transformation” in my opinion the term transformation without some examples is reductive. It is more effectiveness the term cycle, because this term is more understandable and includes nutrient transformation and plant-soil-nutrient enrollment.

Author Response: We agree with reviewer’s opinion and suggestion that the term of “cycle” is more suitable than “transformation” in this sentence. In the revised manuscript, the term “transformation” was replaced by term “cycle” as suggestion. Please see line 16.

Rows 86-87: if you write “studies” you must cite them or show them close to your affirmation. If you referred to Du et al. 2021 it is only one study. Please check if there are more citation about this sentence.

Author Response: We are sorry that we have made this mistake in the manuscript text. In the revised manuscript, relevant references were double-checked and carefully supplemented in the specific place. Please see line 87.

Rows 227-229: It is better to avoid the direct speech in scientific papers, because it looks not professional. Therefore, it will be better to change this sentence into a passive form.

Author Response: Thank you for pointing this out. In the revised manuscript, the sentence was changed into a passive form as suggestion. Please see line 224-225.

Rows 369-372: when you write that “researchers” do something you must put their references, because you are affirming that someone else has done something, and this need to be supported.

Author Response: We are sorry that we have made this mistake in the manuscript text. In the revised version, relevant references were carefully supplemented behind the sentence. Please see line 371-372.

Rows376-379: In my opinion, since you have observed an effect on microbial communities by the different cultivars, it would be desirable that the selection of microbial strains (specific of the tea cultivars), which are linked to nutrient cycles, can be selected and used as "starter" cultures to promote and accelerate growth of new tea plantations Please add some considerations about this effect, both here in the conclusion and in the abstract.

Author Response: Thank you for kind reminding. We agree with reviewer’s opinion and suggestion that some specific microbial strains linked to nutrient cycles should be selected and used as "starter" cultures to promote and accelerate growth of new tea plantations in the study. In the revised manuscript, we supplemented this relevant information in the conclusion section according to reviewer’s suggestion. Please see line 476-478.

Reviewer 2 Report

  1. What is the main question addressed by the research?

The most important aim of the research is to discover the factors affecting microbial community characteristics, including microbial diversity and community composition.

  1. Do you consider the topic original or relevant in the field? Does it address a specific gap in the field?

The topic is original and definitely relevant to the field of microbiology. In the manuscript authors analysed broad spectrum of factors that could affect rhizosphere microorganisms.

  1. What does it add to the subject area compared with other published material?

            The manuscript is interesting and provides some novelty in research areas of microbial community of tea trees rhizosphere.

  1. What specific improvements should the authors consider regarding the methodology? What further controls should be considered?

            Definitely boarder spectrum of plat cultivars should be addressed. The authors did not included the possible relations between different groups of microorganisms, especially fungal and bacterial community. There is also a lack of description of microbial community before the tea plantation was established.

  1. Are the conclusions consistent with the evidence and arguments presented and do they address the main question posed?

            Conclusions are generally well formed. However this sentence is unclear: “Meanwhile, the pathotroph group 473

accounted for the largest proportion of the fungal community in the tea soils under all cultivars, the ZC108 cultivar showed a lower pathotroph abundance, presenting an obstacle due to continuous cropping, even in the short term.”

  1. Are the references appropriate?

            References are well selected and appropriate to the research.

  1. Please include any additional comments on the tables and figures.

            Generally the quality of figures should be improved – not always the data presented in figures is well readable.

8. others

2.3. Line 137-139: “The soil attached to the root surface was collected as the rhizosphere soil, and the collected samples were thoroughly mixed to obtain the composite sample.” – was size/weight was the composite sample?

Figure 2.  – the quality of this figure could be better.

Figure 4, 5. – the quality of this figure could be better (legends).

Author Response

Point-by-point responses to the reviewers' comments

To Reviewer #2:

  1. What is the main question addressed by the research?

The most important aim of the research is to discover the factors affecting microbial community characteristics, including microbial diversity and community composition.

  1. Do you consider the topic original or relevant in the field? Does it address a specific gap in the field?

The topic is original and definitely relevant to the field of microbiology. In the manuscript authors analysed broad spectrum of factors that could affect rhizosphere microorganisms.

  1. What does it add to the subject area compared with other published material?

The manuscript is interesting and provides some novelty in research areas of microbial community of tea trees rhizosphere.

Author Response: Thank you for your comments and suggestions. These comments and suggestions are very helpful for improving the quality of our study.

  1. What specific improvements should the authors consider regarding the methodology? What further controls should be considered?

Definitely boarder spectrum of plant cultivars should be addressed. The authors did not include the possible relations between different groups of microorganisms, especially fungal and bacterial community. There is also a lack of description of microbial community before the tea plantation was established.

Author Response: In the revised version, the reasons for choosing four given varieties was supplemented. Please see line 129. Besides, the relations between bacterial and fungal community composition was analyzed in the supplemental material as Table S1, and supplemented in the result section. Please see line 264-266.

In addition, we also agree with reviewer’s opinions that it will help to improve the overall quality of manuscripts if the information on soil microorganisms before the establishment of tea plantation is provided. Unfortunately, we did not measure the properties of soil microorganisms before the establishment of the tea plantation, resulting in the lack of background values of soil microbial community in the study. In similar studies in the future, we will supplement relevant information.

  1. Are the conclusions consistent with the evidence and arguments presented and do they address the main question posed?

Conclusions are generally well formed. However, this sentence is unclear: “Meanwhile, the pathotroph group accounted for the largest proportion of the fungal community in the tea soils under all cultivars, the ZC108 cultivar showed a lower pathotroph abundance, presenting an obstacle due to continuous cropping, even in the short term.”

Author Response: Thank you for your comments. In our revised manuscript, the unclear sentence of“Meanwhile, the pathotroph group accounted for the largest proportion of the fungal community in the tea soils under all cultivars, the ZC108 cultivar showed a lower pathotroph abundance, presenting an obstacle due to continuous cropping, even in the short term” was revised as “In addition, pathotroph fungi accounted for the largest proportion of the fungal functional guilds under all cultivars, in which ZC108 cultivar showed the lowest pathotroph fungi abundance, indicating that continuous cropping obstacle also exist under tea plant cultivation, even in the short term, and ZC108 cultivar may have strong resistance”. Please see line 472-476.

  1. Are the references appropriate?

References are well selected and appropriate to the research.”

Author Response: Thank you for your comments.

  1. Please include any additional comments on the tables and figures.

Generally, the quality of figures should be improved – not always the data presented in figures is well readable.

Author Response: Thank you for your comments. In our revised manuscript, parts of figures and legends were carefully revised for improving the readability.

  1. others

2.3. Line 137-139: “The soil attached to the root surface was collected as the rhizosphere soil, and the collected samples were thoroughly mixed to obtain the composite sample.” – was size/weight was the composite sample?

Author Response: We are sorry that we did not describe the sentence clearly. In our study, ten tea plants for each variety per plot were chosen, and approximately 5-6 live roots per each tea plant were obtained. Then, the soil attached to the root surface was composed and considered as a composite rhizosphere soil sample (each plot), relevant information was described before this sentence. Please see line 135-138.

Figure 2.  – the quality of this figure could be better.

Author Response: Thank you for pointing this out. In the revised manuscript, Figure 2 was carefully revised for improving the whole quality.

Figure 4, 5. – the quality of this figure could be better (legends).

Author Response: Thank you for pointing this out. In the revised manuscript, the legends of Figure 4 and Figure 5 and were carefully revised for improving the quality of manuscript.
